# Higher Prevalence of Sarcopenia in Knee Osteoarthritis and Its Association with Femoral Intercondylar Cartilage Thickness and Functional Outcomes

**DOI:** 10.3390/life16010004

**Published:** 2025-12-19

**Authors:** Guan-Bo Chen, Chien-Hui Li, Ya-Chun Hu, Yi-Ju Tsai, Ya-Hui Chen, Sheng-Hui Tuan

**Affiliations:** 1Division of Gastroenterology, Department of Internal Medicine, Kaohsiung Armed Forces General Hospital, Kaohsiung 802, Taiwan; tanwein@gmail.com; 2Division of Gastroenterology, Department of Internal Medicine, Tri-Service General Hospital, National Defense Medical Center, Taipei 114, Taiwan; 3Department of Rehabilitation Medicine, Cishan Hospital, Ministry of Health and Welfare, Kaohsiung 842, Taiwan; leechpt100@gmail.com (C.-H.L.); anne0350@gmail.com (Y.-C.H.); 4Institute of Allied Health Sciences, College of Medicine, National Cheng Kung University, Tainan 701, Taiwan; yjtsaincku@gmail.com; 5Department of Physical Therapy, College of Medicine, National Cheng Kung University, Tainan 701, Taiwan; 6Department of Nursing, Mei-Ho University, Pingtung 912, Taiwan; yhchenchis@gmail.com; 7Department of Nursing, Cishan Hospital, Ministry of Health and Welfare, Kaohsiung 842, Taiwan; 8Department of Physical Medicine and Rehabilitation, School of Medicine, College of Medicine, Kaohsiung Medical University, Kaohsiung 807, Taiwan

**Keywords:** knee osteoarthritis, sarcopenia, femoral intercondylar cartilage, musculoskeletal ultrasound, aging

## Abstract

Knee osteoarthritis (KOA) and sarcopenia are prevalent age-related disorders that share common pathophysiological mechanisms such as aging, chronic inflammation, and physical inactivity. Their coexistence may aggravate functional decline and disability. This cross-sectional study aimed to compare the prevalence of sarcopenia between individuals with KOA and matched controls and to explore the relationship between femoral intercondylar cartilage (FIC) thickness and muscle-related parameters. A total of 228 participants (114 KOA, 114 controls) matched by age, sex, and body mass index were enrolled. Assessments included appendicular skeletal muscle mass index (ASMMI), handgrip strength, walking speed, and physical activity. In KOA patients, ultrasound measurements of FIC and quadriceps thickness and the Western Ontario and McMaster Universities Osteoarthritis Index (WOMAC) were additionally obtained. Sarcopenia prevalence was higher in the KOA group than in controls (41.2% vs. 26.3%, *p* = 0.017). Greater FIC thickness was associated with higher ASMMI, stronger handgrip strength, faster walking speed, and lower WOMAC pain and total scores. These findings indicate that FIC thickness may serve as a potential structural biomarker linking cartilage integrity with muscle function in KOA.

## 1. Introduction

Knee osteoarthritis (KOA) is one of the most prevalent degenerative joint diseases worldwide, particularly among older adults, and represents a major contributor to pain, disability, and reduced quality of life [1,2]. In Taiwan, the prevalence is estimated to be approximately 30% among individuals over 50 years of age, with even higher rates observed in those aged ≥65 years [3,4]. KOA is characterized by progressive articular cartilage degeneration, osteophyte formation, joint space narrowing, and synovial inflammation, ultimately leading to pain, stiffness, and functional limitation [5]. Increasing evidence indicates that KOA is a whole-joint disease involving multiple articular tissues. In addition to cartilage loss, pathological changes such as subchondral bone remodeling, meniscal degeneration, and inflammation or fibrosis of the infrapatellar fat pad also contribute to structural deterioration and symptom severity [6].

In addition to its pathological features, the development of KOA is influenced by multiple intrinsic and extrinsic risk factors. Advancing age, female sex, obesity, prior knee injury, joint malalignment, and genetic predisposition are among the most well-established contributors, while occupational or lifestyle factors involving repetitive kneeling, squatting, or heavy loading further increase mechanical stress on the joint [7,8]. These factors contribute to symptom burden and reduced joint function, which may discourage regular physical activity. Consequently, pain and restricted mobility often result in decreased activity levels, accelerating muscle atrophy and further impairing mobility [9,10].

Sarcopenia, defined as a progressive and generalized loss of skeletal muscle mass, strength, and/or physical performance, is increasingly recognized as a public health concern in aging populations [11]. The Asian Working Group for Sarcopenia (AWGS) 2019 consensus specifies that sarcopenia diagnosis requires low muscle mass plus low muscle strength and/or poor physical performance [12]. In Taiwan, the prevalence of sarcopenia, as defined by the AWGS criteria, has been reported to range from 6.7% to 10% among community-dwelling older adults and up to 50.9% among those attending daycare centers [13]. Sarcopenia has been associated with functional decline, falls, fractures, hospitalization, and even mortality [14,15,16].

Recent studies suggest a bidirectional relationship between KOA and sarcopenia [17,18]. Muscle weakness, particularly in the quadriceps, increases knee joint loading and instability, potentially accelerating KOA progression [9,10,19]. Conversely, KOA-related pain and disability can reduce mobility, leading to disuse muscle atrophy [20,21]. Epidemiological have reported that the prevalence of sarcopenia among patients with KOA can be as high as 41.7% [22], which is substantially higher than the prevalence observed in healthy older adults living in the community. Existing evidence indicates that KOA and sarcopenia share common underlying factors such as aging, chronic inflammation, and reduced physical activity [23]. However, the causal direction of this association remains unclear [23,24].

Emerging evidence highlights sarcopenic obesity, a phenotype that combines low muscle mass with excess fat mass, as a particularly detrimental condition in KOA [25,26]. Obesity itself is a well-recognized risk factor for KOA because excess adiposity increases mechanical loading on the joint and promotes systemic inflammation and metabolic dysregulation [27], all of which accelerate cartilage degeneration and impair muscle function [28]. Physical inactivity and chronic inflammation further contribute to both KOA progression and the development of sarcopenia [29,30].

Traditional KOA severity assessment relies on radiographic grading systems such as the Kellgren–Lawrence (K–L) classification [31]. However, radiography lacks sensitivity for early cartilage changes and cannot assess soft tissue alterations involving the synovium, infrapatellar fat pad, or periarticular muscles, nor does it directly evaluate functional implications [32]. Musculoskeletal ultrasound has emerged as a non-invasive, inexpensive, and accessible tool for evaluating articular cartilage and periarticular soft tissues [33]. The femoral intercondylar cartilage (FIC) region, located between the medial and lateral femoral condyles, can be reliably visualized with high-frequency probes, and cartilage thickness may reflect disease severity and joint health [34]. Previous work suggests associations between quadriceps morphology, cartilage integrity, and functional outcomes in KOA [35], but the role of FIC thickness as a quantitative marker for stratifying KOA severity and its link to sarcopenia has not been well established [36]. This study aimed to (1) examine the prevalence of sarcopenia among individuals with and without KOA, (2) evaluate the relationship between FIC thickness and muscle mass, strength, and physical performance, and (3) assess the association between physical activity, nutritional status, and sarcopenia-related indices.

## 2. Materials and Methods

### 2.1. Study Design and Participants

This cross-sectional study recruited rehabilitation outpatients from one tertiary medical center and one local hospital in southern Taiwan. Participants were divided into two groups: the KOA group and the control group. The KOA group consisted of patients who fulfilled the American College of Rheumatology (ACR) clinical classification criteria for KOA, defined as knee pain plus three or more of the following features: age > 50 years, morning stiffness lasting <30 min, crepitus on active motion, bony tenderness, bony enlargement or spur formation, and absence of palpable warmth of the synovium [37]. The control group comprised rehabilitation outpatients without a clinical diagnosis of KOA and without musculoskeletal or neurological disorders of the lower limbs (e.g., lower-limb musculoskeletal pain, low back pain, or lumbosacral radiculopathy) that could influence gait, muscle performance, or physical function. Most individuals in the control group visited the rehabilitation clinic for upper-body or cervical conditions such as shoulder pain, cervical spondylosis, or myofascial pain syndrome. These conditions do not affect lower-limb functional assessments, ensuring that control participants were free from knee-specific or lower-limb impairments that might bias sarcopenia-related evaluations. Eligible participants were required to be over 50 years of age, have sufficient cognitive and physical ability to complete assessments of sarcopenia indicators (body composition, handgrip strength, and gait speed) as well as the Mini Nutritional Assessment, and be able to complete questionnaires on physical activity and dietary frequency; participants who could not independently fill out the questionnaires were included if they could clearly express their responses. Exclusion criteria included a history of lower limb neurological, muscular, or skeletal injury or disease within the previous six months, prior total knee arthroplasty, or long-term use of analgesics for non-degenerative arthritis such as gouty arthritis, rheumatoid arthritis, or psoriatic arthritis.

Sample size was calculated using G*Power 3.1. Given the absence of high-quality studies jointly evaluating sarcopenia and KOA with our outcome set, we based our calculation on a medium effect size. For continuous variables (e.g., appendicular skeletal muscle mass index, ASMMI), a two-sample t test (d = 0.50, α = 0.05, power = 0.80) required 64 participants per group. For prevalence comparison, a chi-square test (w = 0.30) required 72 per group. To account for potential incomplete data, we increased the target sample size by 10%, resulting in a minimum requirement of at least 80 participants per group. The control participants were matched to the KOA group at a 1:1 ratio using propensity scores, with covariates including age, sex, and body mass index (BMI). The study was approved by the Institutional Review Board of National Cheng Kung University Hospital (IRB No. B-ER-111-440) and written informed consent was obtained from all participants prior to data collection.

### 2.2. Outcomes Measured

To minimize selection bias, participants were recruited during four separate four-week periods, with each period including both morning and afternoon outpatient clinic sessions. This approach was designed to capture patients attending clinics at different times and to reduce potential sampling bias related to time-of-day preferences. Because this was a cross-sectional study, all outcomes were assessed at a single time point. The outcome measures comprised both objective and subjective evaluations, as detailed below.

#### 2.2.1. Body Composition

Low skeletal muscle mass is a prerequisite for the diagnosis of sarcopenia. The most widely recognized and accurate tools for its assessment are dual-energy X-ray absorptiometry (DXA) and bioelectrical impedance analysis (BIA). Muscle mass was quantified using appendicular skeletal muscle mass (ASM). The muscle mass index was calculated as ASM divided by height squared (ASM/height^2^, kg/m^2^) in accordance with the Asian Working Group for Sarcopenia (AWGS) 2019 criteria. According to the AWGS criteria, the cutoff values for low muscle mass in men, measured by either DXA or BIA, are ≤7.0 kg/m^2^, whereas in women, the cutoff values are ≤5.4 kg/m^2^ by DEXA and ≤5.7 kg/m^2^ by BIA [12]. We used BIA in this study due to feasibility of implementation and cost-effectiveness. BIA measurements were performed using a multi-frequency InBody S10 analyzer (InBody Co., Seoul, Republic of Korea) with participants in the supine position. This device applies currents across multiple frequencies to estimate segmental impedance and derive fat-free mass, total body water, intracellular water, and extracellular water. Multifrequency BIA has been shown to provide more accurate body composition estimates than traditional single-frequency methods [38].

#### 2.2.2. Quadriceps Muscle Strength

Quadriceps strength of the dominant leg was assessed under standardized positioning using a hand-held dynamometer, the MicroFET3 (Hoggan Health Industries, Salt Lake City, UT, USA). Maximal voluntary isometric contraction (MVIC) of the quadriceps was measured twice, and the mean value of the two trials was recorded as the representative quadriceps strength. For the assessment, participants were seated upright in a chair with a backrest, with their tested foot placed flat on the floor and the knee positioned at 90° flexion. The examiner stabilized the distal third of the participant’s thigh with one hand while placing the dynamometer just proximal to the anterior ankle joint of the tested leg. Participants were then instructed to extend the knee with maximal effort against the resistance applied by the examiner [39]. Each participant performed two trials, separated by a one-minute rest interval, and the mean value was used for subsequent analyses.

#### 2.2.3. Quadriceps Muscle Thickness by Ultrasound

Quadriceps muscle thickness was assessed using a portable ultrasound system (LOGIQ e, General Electric Company, Boston, MA, USA, 2010) equipped with a 12 MHz linear transducer. All measurements were performed under standardized parameters by a physician with extensive clinical experience in musculoskeletal ultrasound (S.H.T.). Previous studies have reported a negative correlation between the severity of KOA and quadriceps thickness, with elderly individuals demonstrating a higher likelihood of quadriceps muscle atrophy compared to healthy controls [34,40].

For the measurement, participants were positioned supine with both lower limbs fully extended and relaxed. The transducer was placed at the midpoint of the line connecting the anterior superior iliac spine and the superior border of the patella. Quadriceps muscle thickness was determined as the sum of the distance between the rectus femoris and the vastus intermedius muscles obtained in transverse plane scanning [41] (Figure 1).

#### 2.2.4. Dominant Handgrip Strength

Several studies have demonstrated that the JAMAR dynamometer provides high test–retest reliability for handgrip strength assessment in older adults [42]. Considering that older individuals may exhibit reduced grip strength and that a more precise scale is required to detect pre- and post-intervention differences, this study employed a JAMAR digital hand dynamometer, which provides readings with a resolution of up to 0.1 lb.

All participants were measured under standard position [42]. One practice trial was provided prior to testing to ensure understanding of the procedure. Each participant then performed three trials, separated by 30 s rest intervals, and the mean value of the three trials was recorded for analysis.

#### 2.2.5. Walking Speed

In accordance with the recommendations of the AWGS [12], usual gait speed was assessed over a 6 m course. Under standard procedure, participants were instructed to walk at their usual pace across a 6 m distance marked on the floor. Timing started upon the verbal cue “ready, go” and stopped when participants crossed the finish line. Each participant completed two trials, with a 10 min rest interval between trials. Walking speed (m/s) was calculated by dividing the distance by the time, and the mean value of the two trials was used for analysis.

#### 2.2.6. Mini Nutritional Assessment

The Mini Nutritional Assessment (MNA) is a comprehensive tool designed to evaluate nutritional status through 18 items, covering dietary intake over the past three months, mobility, and anthropometric measures such as mid-arm circumference and calf circumference. A score of ≥24 indicated a normal nutritional status, 17–23.5 indicated a risk of malnutrition, and <17 indicated malnutrition [43].

#### 2.2.7. International Physical Activity Questionnaire—Self-Administered Short Version

The International Physical Activity Questionnaire—Self-Administered Short Version (IPAQ-SS) was used to assess participants’ self-reported physical activity over the previous 7 days. Activity levels were calculated based on frequency, intensity, and duration of walking, moderate-intensity, and vigorous-intensity activities, and expressed as daily energy expenditure using metabolic equivalents (METs). MET values of 3.3, 4.0, and 8.0 were assigned for walking, moderate-intensity, and vigorous-intensity activities, respectively. Total physical activity was computed as the product of METs, minutes of activity per day, and days per week. Scores were then categorized according to IPAQ classification criteria: (a) low physical activity (<600 MET-min/week), (b) sufficient physical activity (600–3000 MET-min/week), and (c) high physical activity (>3000 MET-min/week) [44].

Since the following outcomes are specific to the assessment of osteoarthritic changes, only participants in the knee OA group underwent the evaluations described in Section 2.2.8, Section 2.2.9 and Section 2.2.10.

#### 2.2.8. Kellgren–Lawrence Grading Scale

Radiographic severity of knee OA was evaluated using the Kellgren–Lawrence grading system (K-L grading), which classifies structural changes into five grades: Grade 0, no radiographic features of OA; Grade 1, doubtful joint space narrowing and possible osteophytic lipping; Grade 2, definite osteophytes with possible narrowing of joint space; Grade 3, moderate multiple osteophytes, definite narrowing of joint space, some sclerosis, and possible deformity of bone ends; and Grade 4, large osteophytes, marked narrowing of joint space, severe sclerosis, and definite deformity of bone ends [31]. The grading was performed by a single experienced physiatrist (S.H.T.).

#### 2.2.9. Western Ontario and McMaster Universities Osteoarthritis Index (WOMAC)

Because the control group consisted of rehabilitation outpatients who frequently presented with non-lower-limb pain, a general visual analog scale (VAS) would not have accurately captured knee-specific symptoms. These participants would likely have difficulty distinguishing knee pain from pain in other regions. Therefore, knee pain was assessed only in the KOA group using the WOMAC. The WOMAC index is a self-administered questionnaire specifically designed to evaluate pain, stiffness, and physical function in patients with hip and knee OA. It consists of 24 items divided into three subscales: pain (5 items), stiffness (2 items), and physical function (17 items). Each item is scored on a 5-point Likert scale ranging from 0 (“none”) to 4 (“extreme”), with a total possible score of 96. Higher scores indicate greater disability [45].

#### 2.2.10. Ultrasonographic Measurement of Femoral Intercondylar Cartilage Thickness

FIC was measured using the same portable ultrasound system described previously for quadriceps assessment, namely the LOGIQ e system (General Electric Company, USA, 2010) equipped with a 12 MHz linear transducer. All measurements were performed using standardized musculoskeletal imaging parameters.

Participants were positioned supine with the knee maximally flexed to fully expose the femoral condylar cartilage. The transducer was placed transversely over the suprapatellar region and oriented perpendicular to the skin surface. Dynamic adjustments of the probe angle were made to optimize visualization of the intercondylar articular cartilage. Once the entire contour of the FIC was clearly visualized, the examiner systematically scanned across the intercondylar region to identify the thickest portion of the cartilage layer [46]. The maximum measurable thickness within the FIC region was recorded, and this value was used for analysis (Figure 2). All measurements were performed by the same experienced operator (S.H.T.) to ensure consistency.

### 2.3. Statistical Analysis

All statistical analyses were performed using SPSS v23.0 (IBM Corp., Armonk, NY, USA; 2015). Continuous variables were expressed as mean ± standard deviation (SD), and categorical variables as frequency and percentage. Data normality was evaluated using the Shapiro–Wilk test. For continuous variables that did not satisfy assumptions of normality or homogeneity of variance, non-parametric Mann–Whitney U-tests would have been considered as alternatives, and chi-square tests would have been applied for categorical variables. In the present study, all continuous variables met the assumptions of normal distribution. Therefore, independent t-tests were used for between-group comparisons of continuous variables, while chi-square tests were applied for categorical variables.

Because age, sex, and BMI are major determinants of both KOA and sarcopenia, we used propensity score matching to minimize baseline confounding, propensity score matching (PSM) was performed using age, sex, and BMI as matching covariates. Propensity scores were estimated using a logistic regression model, and participants were matched in a 1:1 nearest-neighbor fashion without replacement, using a caliper width of 0.2 of the standard deviation of the logit of the propensity score. Balance diagnostics confirmed adequate covariate balance between groups after matching, with no significant residual differences across the matching variables. After confirming balance, the same statistical procedures (independent t-tests for continuous variables and chi-square tests for categorical variables) were used to compare outcomes between the matched groups.

For the KOA subgroup analysis stratified by FIC thickness, PSM could not be applied because both subgroups originated from the same KOA cohort and did not represent independent populations for matching. To address potential confounding in this setting, analysis of covariance (ANCOVA) was performed to adjust for age, sex, and BMI.

Pearson’s correlation coefficients were calculated to examine the associations among weekly physical activity, sarcopenia-related indicators, body composition parameters, and nutritional status. This analysis was also conducted in the KOA subgroup. Results were presented in a three-row format: the upper row reports correlation coefficients, the middle row presents unadjusted p-values, and the lower row displays p-values adjusted for multiple comparisons using the False Discovery Rate (FDR) method with the Benjamini–Hochberg procedure. A two-tailed *p*-value < 0.05 was considered statistically significant.

## 3. Results

### 3.1. Baseline Characteristics of Participants

In total, 136 patients with KOA and 138 controls met the initial inclusion criteria. After applying propensity score matching at a 1:1 ratio, 228 participants were included in the final analysis, comprising 114 patients with KOA and 114 individuals in the control group (Figure 3). Before propensity score matching, the KOA and control groups showed no statistically significant differences in age, sex, BMI, or comorbidities, although a higher proportion of participants in the control group reported a regular exercise habit Appendix A. After matching, the KOA and control groups remained well balanced with respect to age, sex distribution, BMI, dietary habits, nutritional supplement use, comorbidities, smoking status, and alcohol consumption, as presented in Table 1. The only variable that continued to differ significantly between groups was a regular exercise habit. Specifically, a higher proportion of participants in the control group reported engaging in regular exercise compared with the KOA group (24.6% versus 8.8%, *p* = 0.001). All other demographic variables showed no significant differences.

### 3.2. Comparison of Sarcopenia Diagnostic Criteria and Body Composition Between Knee Osteoarthritis and Control Groups

The comparison of sarcopenia diagnostic components and body composition measures is presented in Table 2. Most muscle mass and strength indicators were comparable between groups. Notably, walking speed was lower in the KOA group than in the control group, with a significant difference (0.87 ± 0.26 vs. 0.95 ± 0.27 m/s, *p* = 0.024). The prevalence of sarcopenia was also higher among participants with KOA (41.2% vs. 26.3%, *p* = 0.017). Although the distribution of sarcopenia severity did not reach statistical significance, the KOA group showed a trend toward higher proportions of confirmed and severe sarcopenia.

Body composition parameters, including FFMI, body fat percentage, waist to hip ratio, and nutritional status assessed by the MNA score, showed no significant differences. Weekly physical activity expressed in METs was lower in the KOA group compared with the control group, with a significant difference (2101.58 ± 2692.60 vs. 2950.64 ± 2448.70, *p* = 0.013). The categorical distribution of physical activity levels did not differ between groups.

### 3.3. Correlations Between Physical Activity, Sarcopenia Indicators, and Nutritional Status

Correlation analysis revealed several meaningful associations (Table 3). Weekly physical activity showed a modest positive correlation with ASMMI (r = 0.189, adjusted *p* = 0.023) and a modest negative correlation with body fat percentage (r = −0.222, adjusted *p* = 0.007). ASMMI was positively associated with weekly activity (r = 0.189, adjusted *p* = 0.023), walking speed (r = 0.289, adjusted *p* < 0.001), and FFMI (r = 0.280, adjusted *p* < 0.001). Dominant handgrip strength demonstrated moderate positive correlations with walking speed (r = 0.490, adjusted *p* < 0.001) and FFMI (r = 0.497, adjusted *p* < 0.001), along with a moderate negative correlation with body fat percentage (r = −0.409, adjusted *p* < 0.001). Walking speed was modestly positively correlated with FFMI (r = 0.199, adjusted *p* = 0.015) and modestly negatively correlated with body fat percentage (r = −0.333, adjusted *p* < 0.001). In addition, body fat percentage showed a modest positive association with MNA scores (r = 0.250, adjusted *p* = 0.002). No other significant correlations were identified among physical activity, sarcopenia-related measures, and nutritional status.

### 3.4. Subgroup Analysis of Knee Osteoarthritis Patients by Femoral Intercondylar Cartilage Width

In this study, since most patients with KOA were classified as having mild to moderate severity (Kellgren–Lawrence grade II), to further explore the impact of structural severity on muscle mass and functional outcomes among patients with KOA, participants (n = 114) were stratified into two subgroups based on FIC. Because no standardized or clinically validated reference values for FIC thickness are available, and acknowledging that a consensus-based cutoff would be preferable, we adopted an exploratory, data-driven approach. Based on this criterion, the 114 patients with KOA were divided into two subgroups according to their measured FIC thickness by sonography: group 1, with values below the cohort mean (n = 43), and group 2, with values above the cohort mean (n = 71).

The comparison of demographic, sarcopenia-related, nutritional, and functional characteristics between KOA patients stratified by FIC thickness is summarized in Table 4. Demographic variables, including age, sex distribution, height, body weight, BMI, OA duration, and radiographic Kellgren–Lawrence grade, were comparable between the two groups. In contrast, several muscle-related and functional indicators differed significantly between groups. Patients in group 2 (higher FIC thickness) exhibited greater muscle mass and better physical performance, including higher ASMMI (5.90 ± 1.58 vs. 5.05 ± 1.53 kg/m^2^, *p* = 0.006), stronger handgrip strength (23.28 ± 8.14 vs. 20.30 ± 6.92 kg, *p* = 0.048), faster gait speed (0.92 ± 0.26 vs. 0.80 ± 0.26 m/s, *p* = 0.019), and higher FFMI (16.65 ± 1.79 vs. 15.74 ± 1.34 kg/m^2^, *p* = 0.005). Group 2 also demonstrated better knee-related clinical symptoms, with lower WOMAC pain scores (6.12 ± 4.40 vs. 7.80 ± 4.06, *p* = 0.044) and lower total WOMAC scores (18.05 ± 12.29 vs. 23.70 ± 17.97, *p* = 0.049).

To further confirm whether these differences were independent of major confounders, ANCOVA analyses adjusting for age, sex, and BMI were performed. After adjustment, ASMMI, gait speed, FFMI, WOMAC pain, and WOMAC total scores remained significantly higher (or lower, for pain) in group 2, whereas sonographic thickness of left quadriceps muscle was no longer statistically significant.

### 3.5. Correlations Between Femoral Intercondylar Cartilage Thickness, Sarcopenia Indicators, and WOMAC in Patients with Knee Osteoarthritis

Among patients with KOA, FIC thickness on both sides showed strong positive correlations with sonographic quadriceps thickness (r = 0.751, adjusted *p* < 0.001) and modest positive correlations with ASMMI (right r = 0.373; left r = 0.338; both adjusted *p* < 0.001). FIC thickness was also positively associated with walking speed (right r = 0.196, adjusted *p* = 0.047; left r = 0.261, adjusted *p* = 0.013). No significant correlations were observed between FIC thickness and dominant handgrip strength or any WOMAC subscale (Table 5).

Sonographic quadriceps thickness demonstrated modest positive correlations with ASMMI (r = 0.323, adjusted *p* < 0.001) and walking speed (r = 0.358, adjusted *p* < 0.001). Among sarcopenia-related indicators, dominant handgrip strength was modestly negatively correlated with WOMAC pain (r = −0.277, adjusted *p* = 0.006), WOMAC function (r = −0.336, adjusted *p* < 0.001), and WOMAC total score (r = −0.346, adjusted *p* < 0.001). Walking speed showed similar patterns, with modest negative correlations with WOMAC pain (r = −0.325, adjusted *p* < 0.001), stiffness (r = −0.292, adjusted *p* = 0.002), function (r = −0.428, adjusted *p* < 0.001), and total score (r = −0.455, adjusted *p* < 0.001).

## 4. Discussion

Given the complex nature of KOA and sarcopenia, which share overlapping risk factors and present with considerable heterogeneity, this study specifically focused on muscle mass, muscle strength, functional performance, and cartilage morphology. Our findings showed that patients with KOA had a higher prevalence of sarcopenia and slower walking speed compared with controls, despite comparable muscle mass and handgrip strength. Furthermore, subgroup analyses demonstrated that greater FIC thickness was associated with higher muscle mass, stronger grip strength, faster walking speed, and lower WOMAC scores, suggesting a potential link between cartilage integrity and overall musculoskeletal health.

We found no significant differences in demographic characteristics between the KOA and control groups, except for the habit of regular exercise, indicating that the PSM by age, gender, and BMI was effective. The control group had a higher proportion of regular exercisers, consistent with previous evidence that individuals with KOA tend to engage less in physical activity due to pain and mobility limitations [47]. In our study, patients with KOA also demonstrated lower daily activity levels. Such habitual inactivity may partially mediate the association between KOA and sarcopenia and may accelerate muscle loss, functional decline, and disease progression [48]. Current evidence suggests that appropriate physical activity may help reduce the risk of KOA, although the optimal weekly METs threshold remains uncertain. The American College of Sports Medicine recommends at least 150 min of moderate-intensity aerobic exercise per week (approximately 500–1000 MET-minutes) [49], which helps maintain joint health, strengthen periarticular muscles, and potentially decrease the risk of further joint degeneration [50]. These findings highlight the importance of maintaining an active lifestyle and emphasize the bidirectional relationship between physical activity and joint health.

Previous studies have consistently reported a higher prevalence of sarcopenia among individuals with KOA. A cross-sectional study of 1325 women over 70 years found a greater prevalence of sarcopenia in those with lower-extremity OA compared with those without OA (9.1% vs. 3.5%) [51]. Similarly, a meta-analysis including four cross-sectional studies and 7454 participants reported sarcopenia rates of 45.2% in KOA versus 31.2% in controls [52]. In our study, sarcopenia was likewise more common in the KOA group compared with controls (41.2% vs. 26.0%, *p* = 0.044). The literature suggests a bidirectional relationship whereby low muscle mass and reduced strength increase KOA risk and progression, while pain and limited mobility associated with KOA further accelerate muscle loss, forming a vicious cycle [24,53]. Reduced lean mass is often accompanied by increased adiposity, and adipokine-mediated inflammation may contribute to systemic low-grade inflammation and cartilage degeneration [54]. Although our KOA group showed higher body fat and lower FFMI than controls, these differences did not reach statistical significance.

Among the diagnostic components of sarcopenia, only walking speed differed significantly between groups. Walking speed is a well-established predictor of mobility limitation, disability, and mortality in older adults [55]. In our analyses, walking speed was significantly associated with ASMMI, dominant handgrip strength, and all WOMAC domains in the KOA subgroup, indicating that it reflects both muscle status and functional disability. Collectively, these findings support walking speed as a practical indicator of overall functional capacity in individuals with KOA.

In addition, KOA patients with greater FIC thickness demonstrated higher ASMMI, stronger handgrip strength, faster walking speed, and lower WOMAC pain and total scores. These results suggest that FIC thickness may represent a structural biomarker linking cartilage integrity with muscle function and symptoms. This is consistent with Tuna et al., who reported lower FIC values in sarcopenic individuals and identified both KOA and decreased medial FCT as predictors of sarcopenia [56]. Prior studies have also shown that quadriceps strength is positively correlated with FIC and that reduced muscle strength contributes to cartilage thinning and joint deterioration [57]. Experimental models further confirm this relationship; weakening of the quadriceps via botulinum toxin injection resulted in early cartilage degeneration [58], highlighting muscle weakness as a potential driver of cartilage loss and KOA progression. Experimental evidence reinforces this relationship, as quadriceps weakness induced by botulinum toxin leads to early cartilage degeneration [58]. Conversely, interventions targeting quadriceps strengthening may improve both muscle performance and cartilage thickness [34]. These findings suggest a vicious cycle in which sarcopenia accelerates cartilage loss, while KOA-related pain and cartilage damage exacerbate muscle decline.

Consistent with the findings of Tuna et al. [56], both sonographic FIC thickness and quadriceps thickness were significantly correlated with ASMMI and walking speed, two core diagnostic components of sarcopenia. These findings further support the concept that reduced cartilage thickness and diminished muscle mass or performance may mutually reinforce knee osteoarthritis progression. However, FIC thickness was not significantly correlated with WOMAC pain, stiffness, or function scores, nor was quadriceps thickness correlated with WOMAC outcomes. This discrepancy may be attributable to the multidimensional nature of WOMAC, which reflects functional limitations and symptom burden influenced by numerous determinants beyond cartilage morphology. Pain severity, joint alignment, neuromuscular control, psychosocial factors, and comorbid conditions can all attenuate the direct translation of structural measures into functional performance.

While FIC and quadriceps thickness may represent aspects of structural or musculoskeletal reserve, their influence on self-reported disability is inherently moderated by these multifactorial contributors. This aligns with previous literature showing that the associations between muscle mass or strength and functional outcomes are often weak to modest in magnitude [59]. Given the potential value of FIC as a structural biomarker, we further examined whether the observed differences between FIC subgroups were independent of key demographic factors by ANCOVA adjusting for age, sex, and BMI. Several associations, including those with ASMMI, gait speed, FFMI, and WOMAC pain and total scores, remained statistically significant after adjustment, suggesting that FIC thickness provides information beyond the influence of major confounders.

This study has several limitations. First, although the sample size was adequate for statistical analyses, it remains modest and may limit generalizability. Second, the cross-sectional design precludes causal inference regarding the relationship between KOA and sarcopenia. Longitudinal studies are needed to clarify directionality. Third, radiographic severity was determined using Kellgren–Lawrence grades from clinical records, and not all participants received standardized imaging at the time of evaluation. Fourth, although PSM was applied to balance key demographic variables (age, sex, and BMI), residual confounding cannot be completely excluded. Certain variables, such as regular exercise, were not matched because they may function as mediators, and adjusting for them could introduce over-adjustment or collider bias. PSM also could not be applied in the FIC-based subgroup analysis. Therefore, ANCOVA was used to adjust for age, sex, and BMI, but unmeasured factors may still have influenced results. Fifth, knee-specific pain scores were available only for the KOA group because WOMAC was administered exclusively to individuals with knee pathology. A general VAS was not collected, as many control participants presented with pain in unrelated body regions, which would have compromised specificity. Therefore, pain-related effects on gait speed should be interpreted with caution. Finally, although we evaluated physical activity, nutritional status, and sarcopenia-related measures, other potential confounders—such as inflammatory markers, vitamin D levels, or detailed exercise patterns—were not assessed. Larger longitudinal studies with standardized imaging and broader covariate assessment are warranted to validate and extend these findings.

## 5. Conclusions

In summary, this study highlights the close interplay between KOA and sarcopenia. Patients with KOA exhibited a higher prevalence of sarcopenia and slower walking speed compared with controls, with reduced physical activity potentially mediating this relationship. Greater FIC thickness was associated with higher muscle mass, stronger grip strength, faster walking speed, and lower WOMAC scores among patients with KOA, suggesting its potential as a structural biomarker reflecting joint integrity and overall musculoskeletal health.

These findings support the concept of “sarcopenic osteoarthritis” and highlight the importance of comprehensive clinical evaluation that integrates cartilage morphology, muscle health, and functional performance. In practical terms, our results suggest that incorporating sarcopenia screening and ultrasound-based assessment of FIC thickness may help clinicians identify patients at higher risk of functional decline and guide targeted interventions focusing on muscle strengthening and physical activity. Future longitudinal and interventional studies are needed to clarify causal pathways and determine whether early strategies aimed at preserving muscle mass and enhancing physical activity can mitigate disease progression and improve long-term outcomes in this population.

## Figures and Tables

**Figure 1 life-16-00004-f001:**
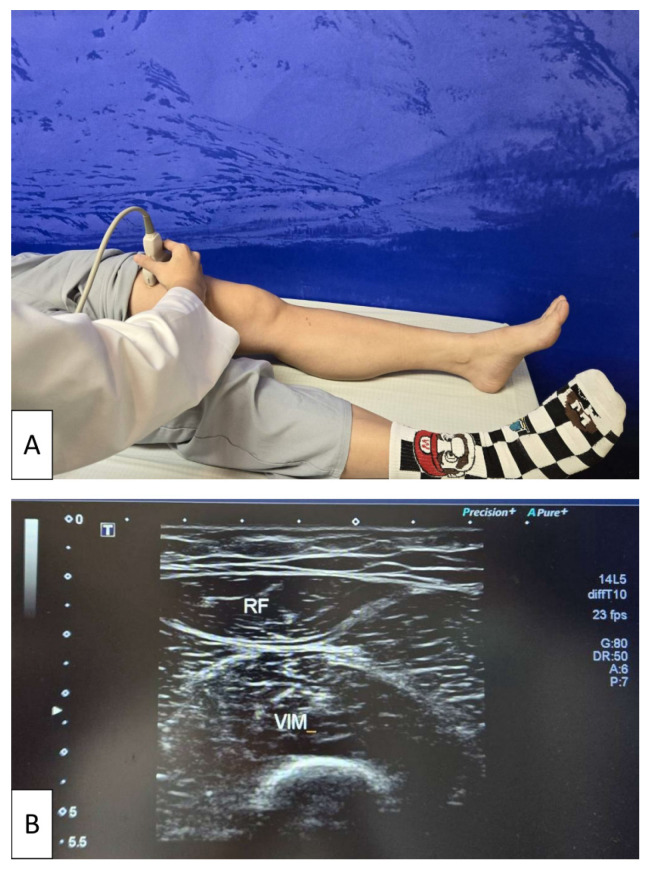
Ultrasonographic assessment of quadriceps muscle thickness. (**A**) Participant positioned supine with both lower limbs fully extended and relaxed while a physician places the linear transducer at the midpoint between the anterior superior iliac spine and the superior border of the patella. (**B**) Representative transverse ultrasound image of the quadriceps, showing measurement of quadriceps muscle thickness as the sum of the distance between the rectus femoris (RF) and vastus intermedius (VIM) muscles. The white arrow on the left side of the image indicates the ultrasound focal depth, which was adjusted to optimize visualization of the muscle layers.

**Figure 2 life-16-00004-f002:**
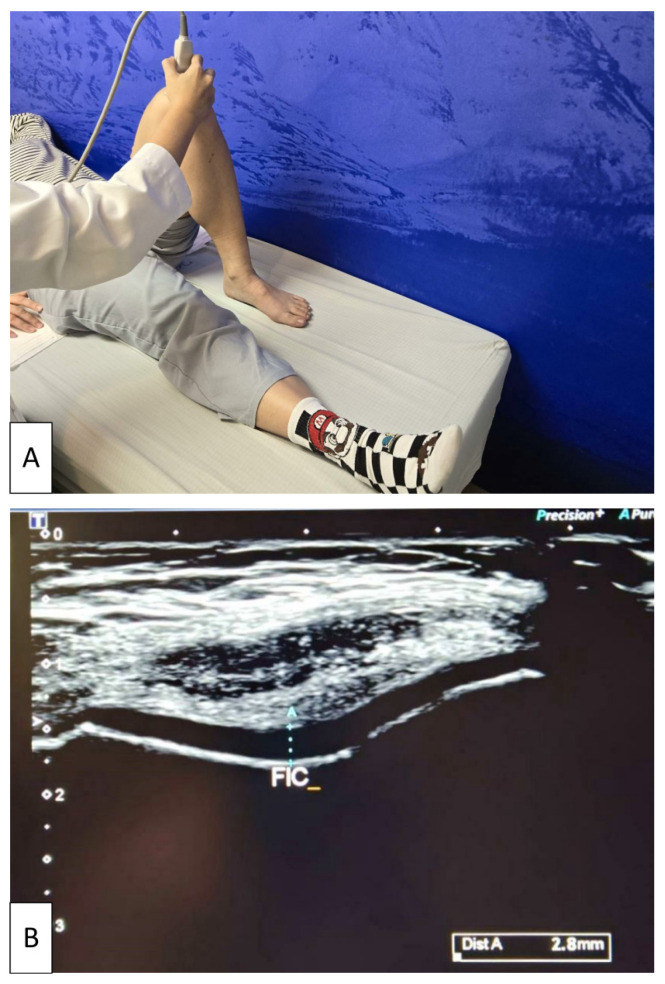
Ultrasonographic measurement of femoral intercondylar cartilage thickness. (**A**) Participant positioned supine with the knee maximally flexed while the examiner places the linear transducer transversely over the suprapatellar region. (**B**) Representative transverse ultrasound image demonstrating measurement of cartilage thickness at the femoral intercondylar notch (FIC). The light blue dotted line indicates the maximum cartilage thickness at the FIC, obtained by dynamically adjusting the probe to optimize visualization of the articular cartilage. The white arrow on the left side of the image represents the ultrasound focal depth setting.

**Figure 3 life-16-00004-f003:**
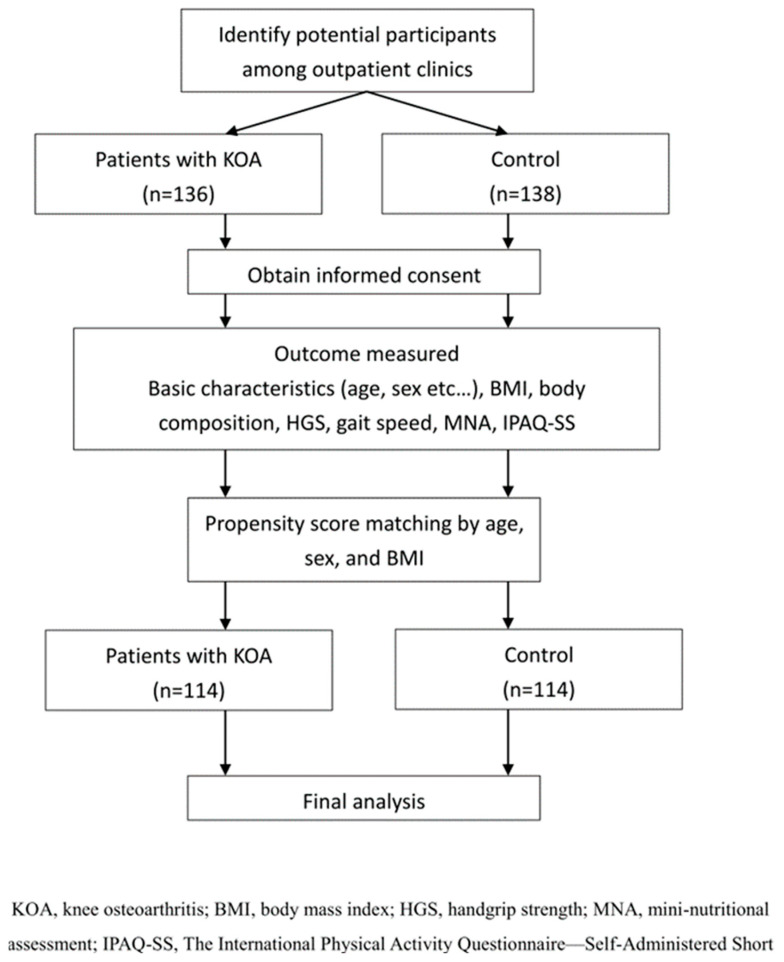
Flow diagram of participant selection and analysis process.

**Table 1 life-16-00004-t001:** Comparison of demographic and baseline characteristics between the knee osteoarthritis and the control group.

Characteristics	Knee OA (n = 114)	Control (n = 114)	*p*-Value
Age (years)	66.50 ± 9.74	66.44 ± 10.93	0.965
Sex, n (%)			0.433
Male	24 (21.1%)	24 (21.1%)	
Female	89 (78.9%)	89 (78.9%)	
Height (cm)	158.72 ± 6.83	158.52 ± 7.23	0.847
Weight (kg)	63.39 ± 10.52	62.17 ± 11.84	0.412
Body mass index (kg/m^2^)	21.58 ± 7.86	20.49 ± 8.29	0.309
Dietary habit			0.806
Non-vegetarian	106 (92.3%)	104 (91.2%)	
Vegetarian	8 (7.7%)	10 (8.8%)	
Nutritional supplement use			0.320
Yes	97 (85.1%)	102 (89.5%)	
No	17 (14.9%)	12 (10.5%)	
Comorbidity			0.741
Yes	92 (80.7%)	90 (78.9%)	
No	22 (19.3%)	24 (21.1%)	
Smoking			0.408
Yes	2 (1.8%)	4 (3.5%)	
No	112 (98.2%)	110 (96.5%)	
Alcohol consumption			0.866
Yes	21 (18.4%)	22 (19.3%)	
No	93 (81.6%)	92 (80.7%)	
Regular exercise habit			0.001 *
Yes	10 (8.8%)	28 (24.6%)	
No	104 (91.2%)	86 (75.4%)	

Values are presented as mean ± SD or number (percentage), * *p* < 0.05.

**Table 2 life-16-00004-t002:** Comparison of Sarcopenia Diagnostic Criteria and Body Composition Between Knee Osteoarthritis and Control Groups.

Variables	Knee OA Group (n = 114)	Control Group (n = 114)	*p*-Value
ASMMI (kg/m^2^)	5.42 ± 1.71	5.59 ± 1.64	0.444
Handgrip Strength (kg)	22.36 ± 7.62	23.10 ± 7.59	0.463
Walking Speed (m/s)	0.87 ± 0.26	0.95 ± 0.27	0.024 *
Sarcopenia			0.017 *
Yes	47 (41.2%)	30 (26.3%)	
No	67 (58.8%)	84 (73.7%)	
Sarcopenia Severity ^a^			0.059
No	20 (17.5%)	30 (26.3%)	
Possible	47 (41.2%)	55 (48.2%)	
Confirmed	25 (21.9%)	18 (15.8%)	
Severe	22 (19.4%)	11 (9.7%)	
FFMI (kg/m^2^)	16.26 ± 1.75	16.37 ± 1.84	0.644
Body Fat Percentage (%)	33.87 ± 8.53	31.84 ± 8.45	0.072
Waist-to-Hip Ratio	1.00 ± 0.88	0.92 ± 0.11	0.337
MNA Score	26.84 ± 2.33	26.85 ± 2.73	0.976
Weekly Physical Activity (METs)	2101.58 ± 2692.60	2950.64 ± 2448.70	0.013 *
Physical Activity Level			0.106
Insufficient (0 < METs < 600)	54 (47.4%)	43 (37.7%)	
Adequate (600 < METs < 3000)	26 (22.8%)	27 (23.7%)	
High (>3000 METs)	34 (29.8%)	44 (38.6%)	

ASMMI: Appendicular Skeletal Muscle Mass Index; FFMI: Fat-Free Mass Index; MNA: Mini Nutritional Assessment; METs: metabolic equivalents. ^a^ Possible sarcopenia: only one of handgrip strength or gait speed below the diagnostic cutoff; Confirmed sarcopenia: either handgrip strength or gait speed below the cutoff, with ASMMI also below the cutoff; Severe sarcopenia: both handgrip strength and gait speed below the cutoff, with ASMMI also below the cutoff. * *p* < 0.05.

**Table 3 life-16-00004-t003:** Correlations between physical activity, sarcopenia indicators, and nutritional status.

		Criteria of Sarcopenia			Body Composition		
	Weekly Activity	ASMMI	Dominant Handgrip Strength	Walking Speed	FFMI	Body Fat	MNA
**Weekly Activity**		0.189 0.013 * 0.023 *	0.120 0.113 0.158	0.012 0.873 0.873	0.026 0.729 0.806	−0.222 0.002 * 0.007 *	0.047 0.537 0.626
**ASMMI**	0.189 0.013 * 0.023 *		0.086 0.255 0.315	0.289 <0.001 * <0.001 *	0.280 <0.001 * <0.001 *	−0.10 0.189 0.249	0.222 0.003 * 0.007 *
**Dominant Handgrip strength**	0.120 0.113 0.158	0.086 0.255 0.315		0.490 <0.001 * <0.001 *	0.497 <0.001 * <0.001 *	−0.409 <0.001 * <0.001 *	−0.020 0.793 0.833
**Walking speed**	0.012 0.873 0.873	0.289 <0.001 * <0.001 *	0.490 <0.001 * <0.001 *		0.199 <0.008 * 0.015 *	−0.333 <0.001 * <0.001 *	−0.164 0.029 * 0.047 *
**FFMI**	0.026 0.729 0.806	0.280 <0.001 * <0.001 *	0.497 <0.001 * <0.001 *	0.199 0.008 * 0.015 *		−0.200 0.008 * 0.015 *	−0.134 0.008 * 0.015 *
**Body fat**	−0.222 0.002 * 0.007 *	−0.10 0.189 0.249	−0.409 <0.001 * <0.001 *	−0.333 <0.001 * <0.001 *	−0.200 0.008 * 0.015 *		0.250 <0.001 * 0.002 *
**MNA**	0.047 0.537 0.626	0.222 0.003 * 0.007 *	−0.020 0.793 0.833	−0.164 0.029 * 0.047 *	−0.134 0.008 * 0.015 *	0.250 <0.001 * 0.002 *	

ASMMI: Appendicular Skeletal Muscle Mass Index; FFMI: Fat-Free Mass Index; MNA: Mini Nutritional Assessment. Data are presented as follows: the upper row shows the correlation coefficient, the middle row indicates the *p* value, and the lower row represents the adjusted *p* value using False Discovery Rate using the Benjamini–Hochberg procedure. * *p* < 0.05.

**Table 4 life-16-00004-t004:** Comparisons of demographic, clinical, and functional characteristics between knee osteoarthritis patients stratified by femoral intercondylar cartilage width.

		Group 1 (n = 43)	Group 2 (n = 71)	*p* Value	Adjusted *p* Value ^a^
**Demographic data**					
Age (years)		69.4 ± 8.93	67.78 ± 8.40	0.188	Not applicable
Gender					
Male		8 (18.60%)	16 (22.54%)	0.793	
Female		35 (81.40%)	55 (77.46%)		
Height (cm)		156.71 ± 5.84	158.3 ± 7.12	0.219	
Body Weight (kg)		60.68 ± 10.98	63.78 ± 10.61	0.126	
BMI (kg/m^2^)		24.72 ± 4.31	25.11 ± 3.71	0.395	
OA duration (years)		5.40 ± 2.60	5.00 ± 2.90	0.581	
Radiological KL grade					
grade 1		4 (9.30%)	7 (9.86%)	0.995	
grade 2		28 (65.12%)	46 (64.79%)		
grade 3		11 (25.58%)	18 (25.35%)		
**Sonographic, sarcopenia indicators, nutritional, and functional data**					
Sonographic-Quadriceps thickness (right)		2.46 ± 0.64	2.60 ± 0.72	0.297	0.461
Sonographic-Quadriceps thickness (left)		2.39 ± 0.56	2.63 ± 0.62	0.049 *	0.317
ASMMI		5.05 ± 1.53	5.90 ± 1.58	0.006 *	<0.001 *
HGS		20.30 ± 6.92	23.28 ± 8.14	0.048 *	0.049 *
Gait speed		0.80 ± 0.26	0.92 ± 0.26	0.019 *	0.012 *
FFMI		15.74 ± 1.34	16.65 ± 1.79	0.005 *	0.007 *
% Body fat		21.93 ± 8.84	21.06 ± 6.94	0.560	0.581
MNA		27.41 ± 1.99	26.62 ± 2.49	0.080	0.071
WOMAC	Pain	7.80 ± 4.06	6.12 ± 4.40	0.044 *	0.042 *
	Stiffness	2.41 ± 1.83	2.06 ± 1.95	0.344	0.926
	Physical function	13.17 ± 10.77	9.58 ± 10.43	0.081	0.734
	Total	23.70 ± 17.97	18.05 ± 12.29	0.049 *	0.043 *

The values are given as mean and the standard deviation or number of patients, with the percentage in parentheses. BMI: body mass index; OA: Osteoarthritis; KL: Kellgren-Lawrence Scale; ASMMI: appendicular skeletal muscle mass index; HGS: handgrip strength; FFMI: fat-free mass index; %Body fat: percentage of fat of the whole body; MNA: mini nutritional assessment; WOMAC: The Western Ontario and McMaster Universities Arthritis Index. Group 1: sonographic thickness of femoral intercondylar cartilage (right < 0.22 mm; left < 0.21 mm). Group 2: sonographic thickness of femoral intercondylar cartilage (right ≥ 0.22 mm; left ≥ 0.21 mm). * *p* < 0.05. ^a^ Differences were further examined using analysis of covariance (ANCOVA) with adjustment for age, sex, and body mass index.

**Table 5 life-16-00004-t005:** Correlations between femoral intercondylar cartilage, sarcopenia indicators, and WOMAC.

				Criteria of Sarcopenia			WOMAC			
	FIC, Right	FIC, Left	Sono-Quad	ASMMI	DHG	Walking Speed	Pain	Stiffness	Function	Total
FIC, right		0.768 <0.001 * <0.001 *	0.751 <0.001 * <0.001 *	0.373 <0.001 * <0.001 *	0.075 0.451 0.503	0.196 0.047 * 0.085	−0.089 0.381 0.455	−0.074 0.461 0.503	−0.058 0.566 0.582	−0.089 0.390 0.455
FIC, left	0.768 <0.001 * <0.001 *		0.742 <0.001 * <0.001 *	0.338 <0.001 * <0.001 *	0.068 0.495 0.524	0.261 0.007 * 0.013 *	−0.087 0.392 0.455	−0.032 0.750 0.750	−0.090 0.365 0.455	−0.088 0.391 0.455
Sono-Quad	0.751 <0.001 * <0.001 *	0.742 <0.001 * <0.001 *		0.323 <0.001 * <0.001 *	0.081 0.501 0.523	0.358 <0.001 * <0.001 *	−0.086 0.367 0.481	−0.058 0.480 0.512	−0.092 0.323 0.418	−0.910 0.395 0.461
ASMMI	0.373 <0.001 * <0.001 *	0.338 <0.001 * <0.001 *	0.323 <0.001 * <0.001 *		0.150 0.096 0.144	0.331 <0.001 * <0.001 *	−0.114 0.230 0.318	−0.153 0.100 0.144	−0.167 0.073 0.125	−0.169 0.078 0.127
DHG	0.075 0.451 0.503	0.068 0.495 0.524	0.081 0.501 0.523	0.150 0.096 0.144		0.461 <0.001 * <0.001 *	−0.277 0.003 * 0.006 *	−0.153 0.100 0.144	−0.336 <0.001 * <0.001 *	−0.346 <0.001 * <0.001 *
Walking speed	0.196 0.047 * 0.085	0.261 0.007 * 0.013 *	0.358 <0.001 * <0.001 *	0.331 <0.001 * <0.001 *	0.461 <0.001 * <0.001 *		−0.325 <0.001 * <0.001 *	−0.292 0.001 0.002 *	−0.428 <0.001 * <0.001 *	−0.455 <0.001 * <0.001 *
WOMAC-Pain	−0.089 0.381 0.455	−0.087 0.392 0.455	−0.086 0.367 0.481	−0.114 0.230 0.318	−0.277 0.003 * 0.006 *	−0.325 <0.001 * <0.001 *		0.434 <0.001 * <0.001 *	0.666 <0.001 <0.001 *	0.810 <0.001 * <0.001 *
WOMAC-Stiffness	−0.074 0.461 0.503	−0.032 0.750 0.750	−0.058 0.480 0.512	−0.153 0.100 0.144	−0.153 0.100 0.144	−0.292 0.001 0.002 *	0.434 <0.001 * <0.001 *		0.596 <0.001 * <0.001 *	0.662 <0.001 * <0.001 *
WOMAC-Function	−0.058 0.566 0.582	−0.090 0.365 0.455	−0.092 0.323 0.418	−0.167 0.073 0.125	−0.336 <0.001 * <0.001 *	−0.428 <0.001 * <0.001 *	0.666 <0.001 <0.001 *	0.596 <0.001 * <0.001 *		0.972 <0.001 * <0.001 *
WOMAC-Total	−0.089 0.390 0.455	−0.088 0.391 0.455	−0.910 0.395 0.461	−0.169 0.078 0.127	−0.346 <0.001 * <0.001 *	−0.455 <0.001 * <0.001 *	0.810 <0.001 * <0.001 *	0.662 <0.001 * <0.001 *	0.972 <0.001 * <0.001 *	

FIC: femoral intercondylar cartilage; Sono-Quad: the maximum quadriceps muscle thickness measured by sonography, taken from either the right or the left side; ASMMI: Appendicular Skeletal Muscle Mass Index; DHG: dominant handgrip strength; WOMAC: The Western Ontario and McMaster Universities Arthritis Index. Data are presented as follows: the upper row shows the correlation coefficient, the middle row indicates the *p* value, and the lower row represents the adjusted *p* value using False Discovery Rate using the Benjamini–Hochberg procedure. * *p* < 0.05.

## Data Availability

The original contributions presented in this study are included in the article/Appendix A. Further inquiries can be directed to the corresponding author (pj73010@hotmail.com).

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
