# Peer review of "Higher Prevalence of Sarcopenia in Knee Osteoarthritis and Its Association with Femoral Intercondylar Cartilage Thickness and Functional Outcomes"

_life, 2025, doi:10.3390/life16010004_

Round 1
Reviewer 1 Report
Comments and Suggestions for Authors
The study is interesting. However, I haves several comments to improve the manuscript.
Introduction
Lines 48-51: It should be specified that KOA is a whole joint disease, involving all joint tissues. Meniscal degeneration and inflammation and fibrosis of the infrapatellar fat pad should be added.
Risk factors for KO are lacking. Please add.
Lines 72-76: please note that obesity is one risk factor for OA (ie DOI: 10.1002/jcp.25716 etc).
Lines 77-79: please add that changes in soft tissues (ie synovial membrane, infrapatellar fat pad and muscles) cannot be evaluated by x-rays.
In the introduction, there is no mention about dietary patterns and OA/sarcopenia but this is reported in the aim of the study. Please clarify.
Methods
Authors reported that this is a cross-sectional study. Please follow STROBE guidelines and submit the checklist as supplementary material.
It is unclear if patients signed the informed consent to participate to the study. This is mandatory.
Section 2.2.1: The authors should specify which exact variables were extracted to quantify muscle mass (e.g., ASM or total lean mass), as well as how the final muscle mass index was calculated (e.g., ASM/height²). In addition, the description lacks essential technical details regarding the measurement tools: the model, brand, manufacturer and paramenters used of the DXA device are missing, and the same information should be reported for the BIA device to ensure full reproducibility.
Lines 165-168: please add these studies.
Figure 1: please modify following the journal guidelines.
Section 2.2.10. Please specify the instrument and parameters used.
Figure 2b: please highlight the FIC.
Lines 283-284: please specify the tests used.
In the statistical analysis, propensity score is not mentioned. Please explain.
Results
Please add a flowchart of patients enrolment.
Please add a table in the supplementary files reporting the demographic data of the initial population.
Why did authors apply PSM? Were the groups different in terms of baseline characteristics (e.g., age, sex, BMI, comorbidities)? Please clarify.
Table 3 and 5 are unclear. R and p-value should be clarified.
In the KOA subgroup stratified by femoral intercondylar cartilage thickness, please clarify whether the observed differences remain significant after adjustment for covariates such as age, sex, and BMI.
Table 4: please correct “Sonogrpahic”.
Some outcomes, such as walking speed and prevalence of sarcopenia, might still be influenced by residual confounding factors (e.g., regular exercise). Consider performing multivariate regression analyses post-matching to assess the independent effects of relevant variables.
Discussion
Lines 432-437: please summarize this part.
Lines 439-440: please add references.
Lines 454-456: please rewrite this sentence.
Other comments:
Please add a space between the square bracket and the final word of the sentence.
Please check references as the numbers are reported twice.
Author Response
We sincerely appreciate the valuable suggestions provided by Reviewer 1. A point-by-point response to each comment has been completed and is included in the attached file for the editor’s review.

Reviewer 2 Report
Comments and Suggestions for Authors
The article "Higher Prevalence of Sarcopenia in Knee Osteoarthritis and Its Association with Femoral Intercondylar Cartilage Thickness and Functional Outcomes" examines the relationship between articular cartilage thickness and disease parameters, as well as body composition, in individuals with knee osteoarthritis. The study design is well-designed and described, the required cohort size was estimated, and the data were qualitatively analyzed. I would like to clarify a few questions before recommending the manuscript for publication:
- What criterion was used to establish the diagnosis of sarcopenia? If based on gait speed, could this be related to the different pain levels in the experimental and control groups? I believe pain levels should be reported not only using the WOMAC questionnaire but also using the visual analog scale in Tables 1 or 2.
- What is the practical value of this study? It certainly expands our understanding of the relationship between OA and body composition, but what can be recommended for practical medicine?
- What was the basis for choosing the cartilage thickness cutoff point (21-22 mm)?
Author Response
We sincerely appreciate the valuable suggestions provided by Reviewer 2. A point-by-point response to each comment has been completed and is included in the attached file for the editor’s review.

Reviewer 3 Report
Comments and Suggestions for Authors
Osteoarthritis involves the degradation of articular cartilage, which impairs joint function. Sarcopenia reduces muscle support and stability around the joint, accelerating cartilage degeneration, while cartilage loss and joint pain lead to decreased activity, worsening sarcopenia. Their association is a bidirectional relationship that significantly impacts joint health and function. Therefore, the study of Dr. Chen et al. seeking the association between FIC thickness and OA traits is very important.
Comments
- Lines 96, 101: The number of patients in each group should be indicated.
- Tables 1-5: All non-statistically significant P-values should be indicated as “n.s.”
- Results: The authors should not repeat the data from Tables in the text of Results section.
- Line 320 vs Table 2: All the typos should be corrected.
- Discussion is wordy. All the repeats (for example, Lines 429 vs 431) should be corrected.
- Lines 499-501: If so, the authors should explain the importance of all their detailed analyses related to FIS thickness.
Author Response
We sincerely appreciate the valuable suggestions provided by Reviewer 3. A point-by-point response to each comment has been completed and is included in the attached file for the editor’s review.

Round 2
Reviewer 1 Report
Comments and Suggestions for Authors
Please note that the caption of the figures should be reported below each figure.